# HetGPT: Harnessing the Power of Prompt Tuning in Pre-Trained Heterogeneous Graph Neural Networks

## ABSTRACT

Graphs have emerged as a natural choice to represent and analyze the intricate patterns and rich information of the Web, enabling applications such as online page classification and social recommendation. The prevailing "*pre-train, fine-tune*" paradigm has been widely adopted in graph machine learning tasks, particularly in scenarios with limited labeled nodes. However, this approach often exhibits a misalignment between the training objectives of pretext tasks and those of downstream tasks. This gap can result in the "negative transfer" problem, wherein the knowledge gained from pre-training adversely affects performance in the downstream tasks. The surge in prompt-based learning within Natural Language Processing (NLP) suggests the potential of adapting a "*pre-train, prompt*" paradigm to graphs as an alternative. However, existing graph prompting techniques are tailored to homogeneous graphs, neglecting the inherent heterogeneity of Web graphs. To bridge this gap, we propose HetGPT, a general post-training prompting framework to improve the predictive performance of pre-trained heterogeneous graph neural networks (HGNNs). The key is the design of a novel prompting function that integrates a virtual class prompt and a heterogeneous feature prompt, with the aim to reformulate downstream tasks to mirror pretext tasks. Moreover, HetGPT introduces a multi-view neighborhood aggregation mechanism, capturing the complex neighborhood structure in heterogeneous graphs. Extensive experiments on three benchmark datasets demonstrate HetGPT's capability to enhance the performance of state-of-the-art HGNNs on semi-supervised node classification.

## 1 INTRODUCTION

The Web, an ever-expanding digital universe, has transformed into an unparalleled data warehouse. Within this intricate web of data, encompassing diverse entities and patterns, graphs have risen as an intuitive representation to encapsulate and examine the Web's multifaceted content, such as academic articles [7], social media interactions [3], chemical molecules [8], and online grocery items [31]. In light of this, graph neural networks (GNNs) have emerged as the state of the art for graph representation learning, which enables a wide range of web-centric applications such as online page classification [25], social recommendation [4], pandemic trends forecasting [21], and dynamic link prediction [32, 33].

A primary challenge in traditional supervised graph machine learning is its heavy reliance on labeled data. Given the magnitude and complexity of the Web, obtaining annotations can be costly and often results in data of low quality. To address this limitation, the "*pre-train, fine-tune*" paradigm has been widely adopted, where GNNs are initially pre-trained with some self-supervised pretext tasks and are then fine-tuned with labeled data for specific downstream tasks. Yet, this paradigm faces the following challenges:

*Under Review at ACM TheWebConf 2024, ,*
2023.

- **(C1)** Fine-tuning methods often overlook the inherent gap between the training objectives of the pretext and the downstream task. For example, while graph pre-training may utilize binary edge classification to draw topologically proximal node embeddings closer, the core of a downstream node classification task would be to ensure nodes with the same class cluster closely. Such misalignment makes the transferred node embeddings suboptimal for downstream tasks, *i.e.,* negative transfer [34, 43]. The challenge arises: *how to reformulate the downstream node classification task to better align with the contrastive pretext task?*
- **(C2)** In semi-supervised node classification, there often exists a scarcity of labeled nodes. This limitation can cause fine-tuned networks to highly overfit these sparse [29] or potentially imbalanced [22] nodes, compromising their ability to generalize to new and unlabeled nodes. The challenge arises: *how to capture and generalize the intricate characteristics of each class in the embedding space to mitigate this overfitting?*
- **(C3)** Given the typically large scale of pre-trained GNNs, the attempt to recalibrate all their parameters during the fine-tuning phase can considerably slow down the rate of training convergence. The challenge arises: *how to introduce only a small number of trainable parameters in the fine-tuning stage while keeping the parameters of the pre-trained network unchanged?*

One potential solution that could partially address these challenges is to adapt the "*pre-train, prompt*" paradigm from natural language processing (NLP) to the graph domain. In NLP, prompt-based learning has effectively generalized pre-trained language models across diverse tasks. For example, a sentiment classification task like "*The WebConf will take place in the scenic city of Singapore in 2024*" can be reframed by appending a specific textual prompt "*I feel so* [MASK]" to the end. It is highly likely that a language model pre-trained on next word prediction will predict "[MASK]" as "*excited*" instead of "*frustrated*", without necessitating extensive fine-tuning. With this methodology, certain downstream tasks can be seamlessly aligned with the pre-training objectives. While few prior work [5, 19, 27–29] has delved into crafting various prompting templates for graphs, their emphasis remains strictly on homogeneous graphs. This narrow focus underscores the last challenge inherent to the heterogeneous graph structures typical of the Web:

- **(C4)** Homogeneous graph prompting techniques typically rely on the pre-trained node embeddings of the target node or the aggregation of its immediate neighbors' embeddings for downstream node classification, which ignores the intricate neighborhood structure inherent to heterogeneous graphs. The challenge arises: *how to leverage the complex heterogeneous neighborhood structure of a node to yield more reliable classification decisions?*

To comprehensively address all four aforementioned challenges, we propose HetGPT, a general post-training prompting framework tailored for heterogeneous graphs. Represented by the acronym

Heterogeneous Graph Prompt Tuning, HetGPT serves as an auxiliary system for HGNNs that have undergone constrastive pre-training. At the core of HetGPT is a novel *graph prompting function* that reformulates the downstream node classification task to align closely with the pretext contrastive task. We begin with the the *virtual class prompt*, which generalizes the intricate characteristics of each class in the embedding space. Then we introduce the *heterogeneous feature prompt*, which acts as a task-specific augmentation to the input graph. This prompt is injected into the feature space and the prompted node features are then passed through the pre-trained HGNN, with all parameters in a frozen state. Furthermore, a *multi-view neighborhood aggregation* mechanism, that encapsulates the complexities of the heterogeneous neighborhood structure, is applied to the target node, generating a node token for classification. Finally, Pairwise similarity comparisons are performed between the node token and the class tokens derived from the virtual class prompt via the contrastive learning objectives established during pre-training, which effectively simulates the process of deriving a classification decision. In summary, our main contributions include:

- To the best of our knowledge, this is the first attempt to adapt the "*pre-train, prompt*" paradigm to heterogeneous graphs.
- We propose HetGPT, a general post-training prompting framework tailored for heterogeneous graphs. By coherently integrating a virtual class prompt, a heterogeneous feature prompt, and a multi-view neighborhood aggregation mechanism, it elegantly bridges the objective gap between pre-training and downstream tasks on heterogeneous graphs.
- Extensive experiments on three benchmark datasets demonstrate HetGPT's capability to enhance the performance of state-of-the-art HGNNs on semi-supervised node classification.

## 2 RELATED WORK

**Heterogeneous graph neural networks.** Recently, there has been a surge in the development of heterogeneous graph neural networks (HGNNs) designed to learn node representations on heterogeneous graphs [20, 35, 40]. For example, HAN [36] introduces hierarchical attention to learn the node-level and semantic-level structures. MAGNN [7] incorporates intermediate nodes along metapaths to encapsulate the rich semantic information inherent in heterogeneous graphs. HetGNN [42] employs random walk to sample node neighbors and utilizes LSTM to fuse heterogeneous features. HGT [11] adopts a transformer-based architecture tailored for web-scale heterogeneous graphs. However, a shared challenge across these models is their dependency on high-quality labeled data for training. In real-world scenarios, obtaining such labeled data can be resource-intensive and sometimes impractical. This has triggered numerous studies to explore pre-training techniques for heterogeneous graphs as an alternative to traditional supervised learning.

**Heterogeneous graph pre-training.** Pre-training techniques have gained significant attention in heterogeneous graph machine learning, especially under the scenario with limited labeled nodes [18, 39]. Heterogeneous graphs, with their complex types of nodes and edges, require specialized pre-training strategies. These can be broadly categorized into generative and contrastive methods. Generative learning in heterogeneous graphs primarily focuses on reconstructing masked segments of the input graph, either in terms of the underlying graph structures or specific node attributes [6, 10, 30]. On the other hand, contrastive learning on heterogeneous graphs aims to refine node representations by magnifying the mutual information of positive pairs while diminishing that of negative pairs. Specifically, representations generated from the same data instance form a positive pair, while those from different instances constitute a negative pair. Some methods emphasizes contrasting node-level representations [13, 14, 37, 41], while another direction contrasts node-level representations with graph-level representations [15, 24, 26]. In general, the efficacy of contrastive methods surpasses that of generative ones [30], making them the default pre-training strategies adopted in this paper.

**Prompt-based learning on graphs.** The recent trend in Natural Language Processing (NLP) has seen a shift from traditional fine-tuning of pre-trained language models (LMs) to a new paradigm: "*pre-train, prompt*" [17]. Instead of fine-tuning LMs through task-specific objective functions, this paradigm reformulates downstream tasks to resemble pre-training tasks by incorporating textual prompts to input texts. This not only bridges the gap between pre-training and downstream tasks but also instigates further research integrating prompting with pre-trained graph neural networks [28]. For example, GPPT [27] and GraphPrompt [19] introduce prompt templates to align the pretext task of link prediction with downstream classification. GPF [5] and VNT-GPPE [29] employ learnable perturbations to the input graph, modulating pre-trained node representations for downstream tasks. However, all these techniques cater exclusively to homogeneous graphs, overlooking the distinct complexities inherent to the heterogeneity in real-world systems.

## 3 PRELIMINARIES

**Definition 1: Heterogeneous graph.** A heterogeneous graph is defined as $\mathcal{G} = \{\mathcal{V}, \mathcal{E}\}$, where $\mathcal{V}$ is the set of nodes and $\mathcal{E}$ is the set of edges. It is associated with a node type mapping function $\phi : \mathcal{V} \rightarrow \mathcal{A}$ and an edge type mapping function $\varphi : \mathcal{E} \rightarrow \mathcal{R}$. $\mathcal{A}$ and $\mathcal{R}$ denote the node type set and edge type set, respectively. For heterogeneous graphs, we require $|\mathcal{A}| + |\mathcal{R}| > 2$. Let $\mathcal{X} = \{X_A \mid A \in \mathcal{A}\}$ be the set of all node feature matrices for different node types. Specifically, $X_A \in \mathbb{R}^{|\mathcal{V}_A| \times d_A}$ is the feature matrix where each row corresponds to a feature vector $x_i^A$ of node $i$ of type $A$. All nodes of type $A$ share the same feature dimension $d_A$, and nodes of different types can have different feature dimensions.

Figure 1(a) illustrates an example heterogeneous graph with three types of nodes: author (A), paper (P), and subject (S), as well as two types of edges: "write" and "belong to".

**Definition 2: Network schema.** The network schema is defined as $\mathcal{S} = (\mathcal{A}, \mathcal{R})$, which can be seen as a meta template for a heterogeneous graph $\mathcal{G}$. Specifically, network schema is a graph defined over the set of node types $\mathcal{A}$, with edges representing relations from the set of edge types $\mathcal{R}$.

Figure 1(b) presents the network schema for a heterogeneous graph. As per the network schema, we learn that a paper is written by an author and that a paper belongs to a subject.

**Definition 3: Metapath.** A metapath $P$ is a path defined by a pattern of node and edge types, denoted as $A_1 \xrightarrow{R_1} A_2 \xrightarrow{R_2} \cdots \xrightarrow{R_l} A_{l+1}$ (abbreviated as $A_1 A_2 \cdots A_{l+1}$), where $A_i \in \mathcal{A}$ and $R_i \in \mathcal{R}$.

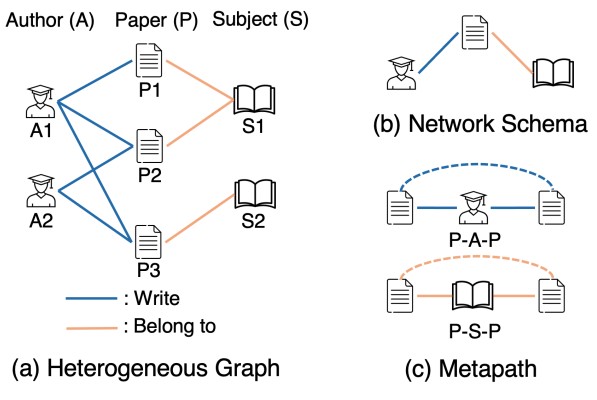

Author (A)  Paper (P)  Subject (S)

(b) Network Schema

: Write

: Belong to

(a) Heterogeneous Graph          (c) Metapath

**Figure 1: A example of a heterogeneous graph.**

Figure 1(c) shows two metapaths for a heterogeneous graph: "PAP" represents that two papers are written by the same author, while "PSP" indicates that two papers share the same subject.

**Definition 4: Semi-supervised node classification.** Given a heterogeneous graph $\mathcal{G} = \{\mathcal{V}, \mathcal{E}\}$ with node features $X$, we aim to predict the labels of the target node set $\mathcal{V}_T$ of type $T \in \mathcal{A}$. Each target node $v \in \mathcal{V}_T$ corresponds to a class label $y_v \in \mathcal{Y}$. Under the semi-supervised learning setting, while the node labels in the labeled set $\mathcal{V}_L \subset \mathcal{V}_T$ are provided, our objective is to predict the labels for nodes in the unlabeled set $\mathcal{V}_U = \mathcal{V}_T \setminus \mathcal{V}_L$.

**Definition 5: Pre-train, fine-tune.** We introduce the "*pre-train, fine-tune*" paradigm for heterogeneous graphs. During the pre-training stage, an encoder $f_\theta$ parameterized by $\theta$ maps each node $v \in \mathcal{V}$ to a low-dimensional representation $\boldsymbol{h}_v \in \mathbb{R}^d$. Typically, $f_\theta$ is an HGNN that takes a heterogeneous graph $\mathcal{G} = \{\mathcal{V}, \mathcal{E}\}$ and its node features $X$ as inputs. For each target node $v \in \mathcal{V}_T$, we construct its positive $\mathcal{P}_v$ and negative sample sets $\mathcal{N}_v$ for contrastive learning. The contrastive head $g_\psi$, parameterized by $\psi$, discriminates the representations between positive and negative pairs. The pre-training objective can be formulated as:

$$\theta^*, \psi^* = \arg\min_{\theta, \psi} \mathcal{L}_{con}\left(g_\psi, f_\theta, \mathcal{V}_T, \mathcal{P}, \mathcal{N}\right), \quad (1)$$

where $\mathcal{L}_{con}$ denotes the contrastive loss. Both $\mathcal{P} = \{\mathcal{P}_v \mid v \in \mathcal{V}_T\}$ and $\mathcal{N} = \{\mathcal{N}_v \mid v \in \mathcal{V}_T\}$ can be nodes or graphs. They may be direct augmentations or distinct views of the corresponding data instances, contingent on the contrastive learning techniques employed.

In the fine-tuning stage, a prediction head $h_\eta$, parameterized by $\eta$, is employed to optimize the learned representations for the downstream node classification task. Given a set of labeled target nodes $\mathcal{V}_L$ and their corresponding label set $\mathcal{Y}$, the fine-tuning objective can be formulated as:

$$\theta^{**}, \eta^* = \arg\min_{\theta^*, \eta} \mathcal{L}_{sup}\left(h_\eta, f_{\theta^*}, \mathcal{V}_L, \mathcal{Y}\right), \quad (2)$$

where $\mathcal{L}_{sup}$ is the supervised loss. Notably, the parameters $\theta$ are initialized with those obtained from the pre-training stage, $\theta^*$.

## 4  METHOD

In this section, we introduce HetGPT, a novel graph prompting technique specifically designed for heterogeneous graphs, to address the four challenges outlined in Section 1. In particular, HetGPT

consists of the following key components: (1) *prompting function design*; (2) *virtual class prompt*; (3) *heterogeneous feature prompt*; (4) *multi-view neighborhood aggregation*; (5) *prompt-based learning and inference*. The overall framework of HetGPT is shown in Figure 2.

### 4.1  Prompting Function Design (C1)

Traditional fine-tuning approaches typically append an additional prediction head and a supervised loss for downstream tasks, as depicted in Equation 2. In contrast, HetGPT pivots towards leveraging and tuning prompts specifically designed for node classification.

In prompt-based learning for NLP, a prompting function employs a pre-defined template to modify the textual input, ensuring its alignment with the input format used during pre-training. Meanwhile, within graph-based pre-training, contrastive learning has overshadowed generative learning, especially in heterogeneous graphs [15, 24, 37], as it offers broader applicability and harnesses overlapping task subspaces, which are optimal for knowledge transfer. Therefore, these findings motivate us to reformulate the downstream node classification task to align with contrastive approaches. Subsequently, a good design of graph prompting function becomes pivotal in matching these contrastive pre-training strategies.

Central to graph contrastive learning is the endeavor to maximize mutual information between node-node or node-graph pairs. In light of this, we propose a graph prompting function, denoted as $l(\cdot)$. This function transforms an input node $v$ into a pairwise template that encompasses a node token $\boldsymbol{z}_v$ and a class token $\boldsymbol{q}_c$:

$$l(v) = [\boldsymbol{z}_v, \boldsymbol{q}_c]. \quad (3)$$

Within the framework, $\boldsymbol{q}_c$ represents a trainable embedding for class $c$ in the downstream node classification task, as explained in Section 4.2. Concurrently, $\boldsymbol{z}_v$ denotes the latent representation of node $v$, derived from the pre-trained HGNN, which will be further discussed in Section 4.3 and Section 4.4.

### 4.2  Virtual Class Prompt (C2)

Instead of relying solely on direct class labels, we propose the concept of a virtual class prompt, a paradigm shift from traditional node classification. Serving as a dynamic proxy for each class, the prompt bridges the gap between the abstract representation of nodes and the concrete class labels they are affiliated with. By leveraging the virtual class prompt, we aim to reformulate downstream node classification as a series of mutual information calculation tasks, thereby refining the granularity and adaptability of the classification predictions. This section delves into the design and intricacies of the virtual class prompt, illustrating how it can be seamlessly integrated into the broader contrastive pre-training framework.

*4.2.1  Class tokens.* We introduce class tokens, the building blocks of the virtual class prompt, which serve as representative symbols for each specific class. Distinct from discrete class labels, these tokens can capture intricate class-specific semantics, providing a richer context for node classification. We formally define the set of class tokens, denoted as $Q$, as follows:

$$Q = \{\boldsymbol{q}_1, \boldsymbol{q}_2, \ldots, \boldsymbol{q}_C\}, \quad (4)$$

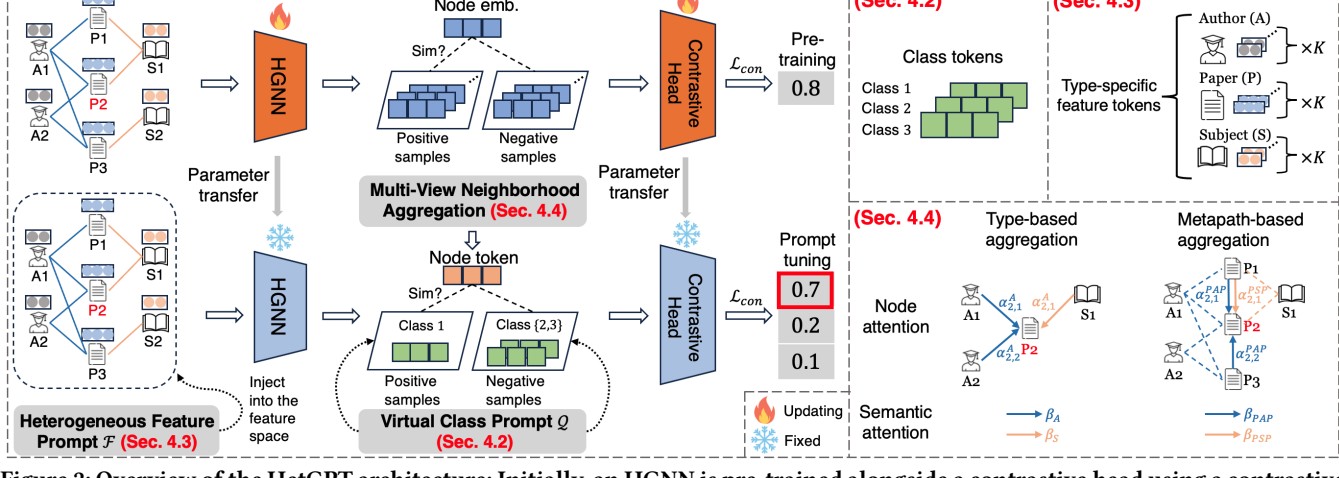

**Figure 2: Overview of the HetGPT architecture:** Initially, an HGNN is pre-trained alongside a contrastive head using a contrastive learning objective, after which their parameters are frozen. Following this, a *heterogeneous feature prompt* (Sec. 4.3) is injected into the input graph's feature space. These prompted node features are then processed by the pre-trained HGNN, producing the prompted node embeddings. Next, a *multi-view neighborhood aggregation* mechanism (Sec. 4.4) captures both local and global heterogeneous neighborhood information of the target node, generating a node token. Finally, pairwise similarity comparisons are performed between this node token and class tokens derived from the *virtual class prompt* (Sec. 4.2) via the same contrastive learning objective from pre-training. As an illustrative example of employing HetGPT for node classification: consider a target node $P_2$ associated with class 1, its positive samples during prompt tuning are constructed using the class token of class 1, while negative samples are drawn from class tokens of classes 2 and 3 (*i.e.,* all remaining classes).

where $C$ is the total number of classes in $\mathcal{Y}$. Each token $q_c \in \mathbb{R}^d$ is a trainable vector and shares the same embedding dimension $d$ with the node representations from the pre-trained network $f_{\theta^*}$.

*4.2.2 Prompt initialization.* Effective initialization of class tokens facilitates a smooth knowledge transfer from pre-trained heterogeneous graphs to the downstream node classification. We initialize each class token, $q_c$, by computing the mean of embeddings for labeled nodes that belong to the respective class. Formally,

$$q_c = \frac{1}{N_c} \sum_{\substack{v \in \mathcal{V}_L \\ y_v = c}} h_v, \quad \forall c \in \{1, 2, \ldots, C\}, \tag{5}$$

where $N_c$ denotes the number of nodes with class $c$ in the labeled set $\mathcal{V}_L$, and $h_v$ represents the pre-trained embedding of node $v$. This initialization aligns each class token with the prevalent patterns of its respective class, enabling efficient prompt tuning afterward.

## 4.3 Heterogeneous Feature Prompt (C3)

Inspired by recent progress with visual prompts in the vision domain [1, 12], we propose a heterogeneous feature prompt. This approach incorporates a small amount of trainable parameters directly into the feature space of the heterogeneous graph $\mathcal{G}$. Throughout the training phase of the downstream task, the parameters of the pre-trained network $f_{\theta^*}$ remain unchanged. The key insight behind this feature prompt lies in its ability to act as task-specific augmentations to the original graph. It implicitly tailors the pre-trained node representations for an effective and efficient transfer of the learned knowledge from pre-training to the downstream task.

Prompting techniques fundamentally revolve around the idea of augmenting the input data to better align with the pretext objectives.

This makes the design of a graph-level transformation an important factor for the efficacy of prompting. To illustrate, let's consider a homogeneous graph $\mathcal{G}$ with its adjacency matrix $A$ and node feature matrix $X$. We introduce $t_\xi$, a graph-level transformation function parameterized by $\xi$, such as changing node features, adding or removing edges, *etc.* Prior research [5, 28] has proved that for any transformation function $t_\xi$, there always exists a corresponding feature prompt $p^*$ that satisfies the following property:

$$f_{\theta^*}(A, X + p^*) \equiv f_{\theta^*}(t_\xi(A, X)) + O_{p\theta}, \tag{6}$$

where $O_{p\theta}$ represents the deviation between the node representations from the graph that's augmented by $t_\xi$ and the graph that's prompted by $p^*$. This discrepancy is primarily contingent on the quality of the learned prompt $p^*$ as the parameters $\theta^*$ of the pre-trained model are fixed. This perspective further implies the feasibility and significance of crafting an effective feature prompt within the graph's input space, which emulates the impact of learning a specialized augmentation function tailored for downstream tasks.

However, in heterogeneous graphs, nodes exhibit diverse attributes based on their types, and each type has unique dimensionalities and underlying semantic meanings. Take a citation network for instance: while paper nodes have features represented by word embeddings derived from their abstracts, author nodes utilize one-hot encoding as features. Given this heterogeneity, the approach used in homogeneous graph prompting methods may not be effective or yield optimal results when applied to heterogeneous graphs, as it uniformly augments node features for all node types via a single and all-encompassing feature prompt.

 

*4.3.1 Type-specific feature tokens.* To address the above challenge, we introduce type-specific feature tokens, which are a set of designated tokens that align with the diverse input features inherent to each node type. Given the diversity in scales and structures across various graphs, equating the number of feature tokens to the node count is often sub-optimal. This inefficiency is especially obvious in large-scale graphs, as this design demands extensive storage due to its $O(|\mathcal{V}|)$ learnable parameters. In light of this, for each node type, we employ a feature prompt consisting of a limited set of independent basis vectors of size $K$, *i.e.*, $\boldsymbol{f}_k^A \in \mathbb{R}^{d_A}$, with $d_A$ as the feature dimension associated with node type $A \in \mathcal{A}$:

$$\mathcal{F} = \{\mathcal{F}_A \mid A \in \mathcal{A}\}, \qquad \mathcal{F}_A = \left\{\boldsymbol{f}_1^A, \boldsymbol{f}_2^A, \ldots, \boldsymbol{f}_K^A\right\}, \qquad (7)$$

where $K$ is a hyperparameter and its value can be adjusted based on the specific dataset in use.

*4.3.2 Prompted node features.* For each node $i$ of type $A \in \mathcal{A}$, its node feature vector $\boldsymbol{x}_i^A$ is augmented by a linear combination of feature token $\boldsymbol{f}_k^A$ through an attention mechanism, where the attention weights are denoted by $w_{i,k}^A$. Consequently, the prompted node feature vector evolves as:

$$\tilde{\boldsymbol{x}}_i^A = \boldsymbol{x}_i^A + \sum_{k=1}^K w_{i,k}^A \cdot \boldsymbol{f}_k^A, \qquad (8)$$

$$w_{i,k}^A = \frac{\exp\left(\sigma\left((\boldsymbol{f}_k^A)^\top \cdot \boldsymbol{x}_i^A\right)\right)}{\sum_{j=1}^K \exp\left(\sigma\left((\boldsymbol{f}_j^A)^\top \cdot \boldsymbol{x}_i^A\right)\right)}, \qquad (9)$$

where $\sigma(\cdot)$ represents a non-linear activation function. Subsequently, we utilize these prompted node features, represented as $\tilde{\mathcal{X}}$, together with the heterogeneous graph, $\mathcal{G}$. They are then passed through the pre-trained HGNN $f_{\theta^*}$ during the prompt tuning phase to obtain a prompted node embedding matrix $\tilde{\boldsymbol{H}}$:

$$\tilde{\boldsymbol{H}} = f_{\theta^*}(\mathcal{G}, \tilde{\mathcal{X}}) \in \mathbb{R}^{|\mathcal{V}| \times d}. \qquad (10)$$

## 4.4 Multi-View Neighborhood Aggregation (C4)

In prompt-based learning for homogeneous graphs, the node token $\boldsymbol{z}_v$ in Equation 3 for a given node $v \in \mathcal{V}$ is directly equated to $\boldsymbol{h}_v$, which is the embedding generated by the pre-trained network $f_{\theta^*}$ [38]. Alternatively, it can also be derived from an aggregation of the embeddings of its immediate neighboring nodes [27]. However, in heterogeneous graphs, such aggregations are complicated due to the inherent heterogeneity of neighboring structures. For example, given a target node with the type "paper", connections can be established either with other "paper" nodes through different metapaths (*e.g.*, PAP, PSP) or with nodes of varied types (*i.e.*, author or subject) based on the network schema. Furthermore, it is also vital to leverage the prompted pre-trained node embeddings $\tilde{\boldsymbol{H}}$ (as detailed in Section 4.3) in the aggregation. Taking all these into consideration, we introduce a multi-view neighborhood aggregation mechanism. This strategy incorporates both type-based and metapath-based neighbors, ensuring a comprehensive representation that captures both local (*i.e.*, network schema) and global (*i.e.*, metapath) patterns.

*4.4.1 Type-based aggregation.* Based on the network schema outlined in Definition 2, a target node $i \in \mathcal{V}_T$ can directly connect to $M$ different node types $\{A_1, A_2, \ldots, A_M\}$. Given the variability

in contributions from different nodes of the same type to node $i$ and the diverse influence from various types of neighbors, we utilize a two-level attention mechanism [36] to aggregate the local information of node $i$. For the first level, the information $\boldsymbol{h}_i^{A_m}$ is fused from the neighbor set $\mathcal{N}_i^{A_m}$ for node $i$ using node attention:

$$\boldsymbol{h}_i^{A_m} = \sigma\left(\sum_{j \in \mathcal{N}_i^{A_m} \cup \{i\}} \alpha_{i,j}^{A_m} \cdot \tilde{\boldsymbol{h}}_j\right), \qquad (11)$$

$$\alpha_{i,j}^{A_m} = \frac{\exp\left(\sigma\left(\mathbf{a}_{A_m}^\top \cdot [\tilde{\boldsymbol{h}}_i \| \tilde{\boldsymbol{h}}_j]\right)\right)}{\sum_{k \in \mathcal{N}_i^{A_m} \cup \{i\}} \exp\left(\sigma\left(\mathbf{a}_{A_m}^\top \cdot [\tilde{\boldsymbol{h}}_i \| \tilde{\boldsymbol{h}}_k]\right)\right)}, \qquad (12)$$

where $\sigma(\cdot)$ is a non-linear activation function, $\|$ denotes concatenation, and $\mathbf{a}_{A_m} \in \mathbb{R}^{2d \times 1}$ is the node attention vector shared across all nodes of type $A_m$. For the second level, the type-based embedding of node $i$, denoted as $\boldsymbol{z}_i^{\text{TP}}$, is derived by synthesizing all type representations $\{\boldsymbol{h}_i^{A_1}, \boldsymbol{h}_i^{A_2}, \ldots, \boldsymbol{h}_i^{A_M}\}$ through semantic attention:

$$\boldsymbol{z}_i^{\text{TP}} = \sum_{i=1}^M \beta_{A_m} \cdot \boldsymbol{h}_i^{A_m}, \quad \beta_{A_m} = \frac{\exp(w_{A_m})}{\sum_{k=1}^M \exp(w_{A_k})}, \qquad (13)$$

$$w_{A_m} = \frac{1}{|\mathcal{V}_T|} \sum_{i \in \mathcal{V}_T} \mathbf{a}_{\text{TP}}^\top \cdot \tanh(\boldsymbol{W}_{\text{TP}} \cdot \boldsymbol{h}_i^{A_m} + \boldsymbol{b}_{\text{TP}}), \qquad (14)$$

where $\mathbf{a}_{\text{TP}} \in \mathbb{R}^{d \times 1}$ is the type-based semantic attention vector shared across all node types, $\boldsymbol{W}_{\text{TP}} \in \mathbb{R}^{d \times d}$ is the weight matrix, and $\boldsymbol{b}_{\text{TP}} \in \mathbb{R}^{d \times 1}$ is the bias vector.

*4.4.2 Metapath-based aggregation.* In contrast to type-based aggregation, metapath-based aggregation provides a perspective to capture global information of a target node $i \in \mathcal{V}_T$. This is attributed to the nature of metapaths, which encompass connections that are at least two hops away. Given a set of defined metapaths $\{P_1, P_2, \ldots, P_N\}$, the information from neighbors of node $i$ connected through metapath $P_n$ is aggregated via node attention:

$$\boldsymbol{h}_i^{P_n} = \sigma\left(\sum_{j \in \mathcal{N}_i^{P_n} \cup \{i\}} \alpha_{i,j}^{P_n} \cdot \tilde{\boldsymbol{h}}_i\right), \qquad (15)$$

$$\alpha_{i,j}^{P_n} = \frac{\exp\left(\sigma\left(\mathbf{a}_{P_n}^\top \cdot [\tilde{\boldsymbol{h}}_i \| \tilde{\boldsymbol{h}}_j]\right)\right)}{\sum_{k \in \mathcal{N}_i^{P_n} \cup \{i\}} \exp\left(\sigma\left(\mathbf{a}_{P_n}^\top \cdot [\tilde{\boldsymbol{h}}_i \| \tilde{\boldsymbol{h}}_k]\right)\right)}, \qquad (16)$$

where $\mathbf{a}_{P_n} \in \mathbb{R}^{2d \times 1}$ is the node attention vector shared across all nodes connected through metapath $P_n$. To compile the global structural information from various metapaths, we fuse the node embeddings $\{\boldsymbol{h}_i^{P_1}, \boldsymbol{h}_i^{P_2}, \ldots, \boldsymbol{h}_i^{P_N}\}$ derived from each metapath into a single embedding using semantic attention:

$$\boldsymbol{z}_i^{\text{MP}} = \sum_{i=1}^N \beta_{P_n} \cdot \boldsymbol{h}_i^{P_n}, \quad \beta_{P_n} = \frac{\exp(w_{P_n})}{\sum_{k=1}^N \exp(w_{P_k})}, \qquad (17)$$

$$w_{P_n} = \frac{1}{|\mathcal{V}_T|} \sum_{i \in \mathcal{V}_T} \mathbf{a}_{\text{MP}}^\top \cdot \tanh(\boldsymbol{W}_{\text{MP}} \cdot \boldsymbol{h}_i^{P_n} + \boldsymbol{b}_{\text{MP}}), \qquad (18)$$

where $\mathbf{a}_{\text{MP}} \in \mathbb{R}^{d \times 1}$ is the metapath-based semantic-attention vector shared across all metapaths, $\boldsymbol{W}_{\text{MP}} \in \mathbb{R}^{d \times d}$ is the weight matrix,

and $\boldsymbol{b}_{\text{MP}} \in \mathbb{R}^{d \times 1}$ is the bias vector. Integrating the information from both aggregation views, we obtain the final node token, $z_i$, by concatenating the type-based and the metapath-based embedding:

$$z_i = \sigma\left(\boldsymbol{W}[z_i^{\text{MP}} \| z_i^{\text{TP}}] + \boldsymbol{b}\right), \tag{19}$$

where $\sigma(\cdot)$ is a non-linear activation function, $\boldsymbol{W} \in \mathbb{R}^{2d \times d}$ is the weight matrix, and $\boldsymbol{b} \in \mathbb{R}^{d \times 1}$ is the bias vector.

## 4.5 Prompt-Based Learning and Inference

Building upon our prompt design detailed in the preceding sections, we present a comprehensive overview of the prompt-based learning and inference process for semi-supervised node classification. This methodology encompasses three primary stages: (1) *prompt addition*, (2) *prompt tuning*, and (3) *prompt-assisted prediction*.

*4.5.1 Prompt addition.* Based on the graph prompting function $l(\cdot)$ outlined in Equation (3), we parameterize it using the trainable virtual class prompt $\boldsymbol{Q}$ and the heterogeneous feature prompt $\mathcal{F}$. To ensure compatibility during the contrastive loss calculation, which we detail later, we use a single-layer Multilayer Perceptron (MLP) to project both $z_v$ and $\boldsymbol{q}_c$, onto the same embedding space. Formally:

$$z_v' = \text{MLP}(z_v), \quad \boldsymbol{q}_c' = \text{MLP}(\boldsymbol{q}_c), \quad l_{Q,\mathcal{F}}(v) = [z_v', \boldsymbol{q}_c']. \tag{20}$$

*4.5.2 Prompt tuning.* Our prompt design allows us to reuse the contrastive head from Equation 1 for downstream node classification without introducing a new prediction head. Thus, the original positive $\mathcal{P}_v$ and negative samples $\mathcal{N}_v$ of a labeled node $v \in \mathcal{V}_L$ used during pre-training are replaced with the virtual class prompt corresponding to its given class label $y_v$:

$$\mathcal{P}_v = \left\{\boldsymbol{q}_{y_v}\right\}, \qquad \mathcal{N}_v = \boldsymbol{Q} \setminus \left\{\boldsymbol{q}_{y_v}\right\}, \tag{21}$$

Consistent with the contrastive pre-training phase, we employ the InfoNCE [23] loss to replace the supervised classification loss $\mathcal{L}_{sup}$:

$$\mathcal{L}_{con} = -\sum_{v \in \mathcal{V}_L} \log\left(\frac{\exp(\text{sim}(z_v', \boldsymbol{q}_{y_v}')/\tau)}{\sum_{c=1}^{C} \exp(\text{sim}(z_v', \boldsymbol{q}_c')/\tau)}\right). \tag{22}$$

Here, $\text{sim}(\cdot)$ denotes a similarity function between two vectors, and $\tau$ denotes a temperature hyperparameter. To obtain the optimal prompts, we utilize the following prompt tuning objective:

$$\boldsymbol{Q}^*, \mathcal{F}^* = \arg\min_{\boldsymbol{Q}, \mathcal{F}} \mathcal{L}_{con}\left(g_{\psi^*}, f_{\theta^*}, l_{Q,\mathcal{F}}, \mathcal{V}_L\right) + \lambda \mathcal{L}_{orth}, \tag{23}$$

where $\lambda$ is a regularization hyperparameter. The orthogonal regularization [2] loss $\mathcal{L}_{orth}$ is defined to ensure the label tokens in the virtual class prompt remain orthogonal during prompt tuning, fostering diversified representations of different classes:

$$\mathcal{L}_{orth} = \left\|\boldsymbol{Q}\boldsymbol{Q}^\top - \boldsymbol{I}\right\|_F^2, \tag{24}$$

where $\boldsymbol{Q} = [\boldsymbol{q}_1, \boldsymbol{q}_2, \dots, \boldsymbol{q}_C]^\top \in \mathbb{R}^{C \times d}$ is the matrix form of the virtual class prompt $\boldsymbol{Q}$, and $\boldsymbol{I} \in \mathbb{R}^{C \times C}$ is an identity matrix.

*4.5.3 Prompt-assisted prediction.* During the inference phase, for an unlabeled target node $v \in \mathcal{V}_U$, the predicted probability of node $v$ belonging to class $c$ is given by:

$$P(y_v = c) = \frac{\exp(\text{sim}(z_v', \boldsymbol{q}_c'))}{\sum_{k=1}^{C} \exp(\text{sim}(z_v', \boldsymbol{q}_k'))}. \tag{25}$$

**Table 1: Detailed statistics of the benchmark datasets. Underlined node types are the target nodes for classification.**

| Dataset | # Nodes | # Edges | Metapaths | # Classes |
|---------|---------|---------|-----------|-----------|
| ACM | Paper: 4,019 Author: 7,167 Subject: 60 | P-A: 13,407 P-S: 4,019 | PAP PSP | 3 |
| DBLP | Author: 4,057 Paper: 14,328 Term: 7,723 Conference: 20 | P-A: 19,645 P-T: 85,810 P-C: 14,328 | APA APCPA APTPA | 4 |
| IMDB | Movie: 4,278 Director: 2,081 Actor: 5,257 | M-D: 4,278 M-A: 12,828 | MAM MDM | 3 |

This equation computes the similarity between the projected node token $z_v'$ and each projected class token $\boldsymbol{q}_c'$, using the softmax function to obtain class probabilities. The class with the maximum likelihood for node $v$ is designated as the predicted class $\hat{y}_v$:

$$\hat{y}_v = \arg\max_c P(y_v = c), \tag{26}$$

## 5 EXPERIMENTS

In this section, we conduct a thorough evaluation of our proposed HetGPT to address the following research questions:

- **(RQ1)** Can HetGPT improve the performance of pre-trained heterogeneous graph neural networks on the semi-supervised node classification task?
- **(RQ2)** How does HetGPT perform under different settings, *i.e.*, ablated models and hyperparameters?
- **(RQ3)** How does the prompt tuning efficiency of HetGPT compare to its fine-tuning counterpart?
- **(RQ4)** How interpretable is the learned prompt in HetGPT?

## 5.1 Experiment Settings

*5.1.1 Datasets.* We evaluate our methods using three benchmark datasets: ACM [44], DBLP [7], and IMDB [7]. Detailed statistics and descriptions of these datasets can be found in Table 1. For the semi-supervised node classification task, we randomly select 1, 5, 20, 40, or 60 labeled nodes per class as our training set. Additionally, we set aside 1,000 nodes for validation and another 1,000 nodes for testing. Our evaluation metrics include Macro-F1 and Micro-F1.

*5.1.2 Baseline models.* We compare our approach against methods belonging to three different categories:

- **Supervised HGNNs:** HAN [36], HGT [11], MAGNN [7];
- **HGNNs with "*pre-train, fine-tune*":**
  - **Generative:** HGMAE [30];
  - **Contrastive (our focus):** DMGI [24], HeCo [37], HDMI [15];
- **GNNs with "*pre-train, prompt*":** GPPT [27].

*5.1.3 Implementation details.* For the homogeneous method GPPT, we evaluate using all the metapaths and present the results with the best performance. Regarding the parameters of other baselines, we adhere to the configuration specified in their original papers.

In our HetGPT model, the heterogeneous feature prompt is initialized using Kaiming initialization [9]. During the prompt tuning phase, we employ the Adam optimizer [16] and search within a

**Table 2: Experiments results on three semi-supervised node classification benchmark datasets. We report the average performance for 10 repetitions. The best results are highlighted in bold, while improved results attributed to HetGPT are underlined. The "+" symbol indicates the integration of HetGPT with the corresponding original models as an auxiliary system.**

| Dataset | Metric | # Train | HAN | HGT | MAGNN | HGMAE | GPPT | DMGI | +HetGPT | HeCo | +HetGPT | HDMI | +HetGPT |
|---|---|---|---|---|---|---|---|---|---|---|---|---|---|
| ACM | Ma-F1 | 1 | $27.08_{\pm2.05}$ | $49.74_{\pm9.38}$ | $38.62_{\pm2.87}$ | $28.00_{\pm7.21}$ | $21.85_{\pm1.09}$ | $47.28_{\pm0.23}$ | $\underline{52.07}_{\pm3.28}$ | $54.24_{\pm8.42}$ | $\underline{55.90}_{\pm8.42}$ | $65.58_{\pm7.45}$ | $\mathbf{71.00}_{\pm5.32}$ |
| | | 5 | $84.84_{\pm0.95}$ | $84.40_{\pm7.48}$ | $84.45_{\pm0.79}$ | $87.34_{\pm1.62}$ | $71.77_{\pm6.73}$ | $86.12_{\pm0.45}$ | $\underline{87.91}_{\pm0.77}$ | $86.55_{\pm1.36}$ | $\underline{87.03}_{\pm1.15}$ | $88.88_{\pm1.73}$ | $\mathbf{91.08}_{\pm0.37}$ |
| | | 20 | $84.37_{\pm1.25}$ | $84.40_{\pm5.31}$ | $85.13_{\pm1.58}$ | $88.61_{\pm1.10}$ | $80.90_{\pm0.88}$ | $86.64_{\pm0.65}$ | $\underline{88.65}_{\pm0.81}$ | $88.09_{\pm1.21}$ | $\underline{88.63}_{\pm0.88}$ | $90.76_{\pm0.79}$ | $\mathbf{92.15}_{\pm0.25}$ |
| | | 40 | $86.33_{\pm0.66}$ | $86.17_{\pm6.26}$ | $86.26_{\pm0.67}$ | $88.31_{\pm1.09}$ | $81.78_{\pm1.46}$ | $87.52_{\pm0.46}$ | $\underline{87.88}_{\pm0.69}$ | $87.03_{\pm1.40}$ | $86.88_{\pm0.95}$ | $90.62_{\pm0.21}$ | $\mathbf{91.31}_{\pm0.39}$ |
| | | 60 | $86.31_{\pm2.16}$ | $86.15_{\pm6.05}$ | $86.56_{\pm1.96}$ | $88.81_{\pm0.72}$ | $84.15_{\pm0.47}$ | $88.71_{\pm0.59}$ | $\underline{90.33}_{\pm0.41}$ | $88.95_{\pm0.85}$ | $\underline{89.13}_{\pm0.59}$ | $91.29_{\pm0.57}$ | $\mathbf{92.09}_{\pm0.35}$ |
| | Mi-F1 | 1 | $49.76_{\pm0.35}$ | $58.52_{\pm6.75}$ | $51.27_{\pm0.45}$ | $40.82_{\pm7.26}$ | $34.32_{\pm3.87}$ | $49.63_{\pm0.25}$ | $\underline{54.29}_{\pm4.49}$ | $54.81_{\pm9.88}$ | $\underline{63.01}_{\pm9.61}$ | $64.89_{\pm8.20}$ | $\mathbf{73.41}_{\pm2.51}$ |
| | | 5 | $84.96_{\pm1.12}$ | $85.11_{\pm4.06}$ | $85.31_{\pm1.14}$ | $87.47_{\pm1.53}$ | $75.41_{\pm3.66}$ | $86.16_{\pm0.47}$ | $\underline{88.05}_{\pm0.77}$ | $86.85_{\pm1.33}$ | $\underline{87.26}_{\pm1.09}$ | $89.01_{\pm1.69}$ | $\mathbf{91.09}_{\pm0.37}$ |
| | | 20 | $83.33_{\pm1.58}$ | $83.05_{\pm3.62}$ | $83.88_{\pm1.60}$ | $88.31_{\pm1.15}$ | $81.20_{\pm0.63}$ | $85.94_{\pm0.64}$ | $\underline{88.40}_{\pm0.79}$ | $87.87_{\pm1.24}$ | $\underline{88.60}_{\pm0.79}$ | $90.55_{\pm0.82}$ | $\mathbf{91.85}_{\pm0.26}$ |
| | | 40 | $86.24_{\pm0.67}$ | $86.21_{\pm3.68}$ | $86.39_{\pm0.69}$ | $88.29_{\pm1.04}$ | $82.02_{\pm1.49}$ | $87.09_{\pm0.47}$ | $\underline{87.78}_{\pm0.79}$ | $86.56_{\pm1.56}$ | $\underline{86.64}_{\pm1.05}$ | $90.41_{\pm0.23}$ | $\mathbf{91.11}_{\pm0.39}$ |
| | | 60 | $85.56_{\pm2.48}$ | $85.49_{\pm4.74}$ | $86.03_{\pm2.40}$ | $88.59_{\pm0.71}$ | $84.16_{\pm0.45}$ | $88.34_{\pm0.63}$ | $\underline{90.13}_{\pm0.43}$ | $88.48_{\pm0.94}$ | $\underline{88.91}_{\pm0.62}$ | $91.16_{\pm0.56}$ | $\mathbf{91.94}_{\pm0.33}$ |
| DBLP | Ma-F1 | 1 | $50.28_{\pm8.41}$ | $70.86_{\pm6.82}$ | $52.52_{\pm8.67}$ | $82.75_{\pm7.96}$ | $39.17_{\pm1.25}$ | $76.00_{\pm3.27}$ | $\underline{81.33}_{\pm1.90}$ | $88.79_{\pm0.44}$ | $\underline{89.44}_{\pm0.54}$ | $88.28_{\pm0.58}$ | $\mathbf{90.25}_{\pm0.29}$ |
| | | 5 | $82.85_{\pm8.60}$ | $82.70_{\pm5.28}$ | $82.24_{\pm0.85}$ | $83.47_{\pm4.57}$ | $54.13_{\pm1.06}$ | $81.12_{\pm1.20}$ | $\underline{81.85}_{\pm1.89}$ | $91.56_{\pm0.23}$ | $\mathbf{91.87}_{\pm0.43}$ | $91.00_{\pm0.38}$ | $\underline{91.39}_{\pm0.46}$ |
| | | 20 | $89.41_{\pm0.61}$ | $89.61_{\pm5.70}$ | $89.36_{\pm0.58}$ | $89.31_{\pm1.47}$ | $71.06_{\pm0.31}$ | $84.03_{\pm1.20}$ | $\underline{84.41}_{\pm1.32}$ | $89.90_{\pm0.37}$ | $\underline{91.17}_{\pm0.52}$ | $91.30_{\pm0.17}$ | $\mathbf{91.64}_{\pm0.31}$ |
| | | 40 | $89.25_{\pm0.55}$ | $89.59_{\pm6.69}$ | $89.42_{\pm0.53}$ | $89.99_{\pm0.45}$ | $73.39_{\pm0.59}$ | $85.43_{\pm1.09}$ | $\underline{85.91}_{\pm0.91}$ | $90.45_{\pm0.31}$ | $\underline{91.48}_{\pm0.41}$ | $90.77_{\pm0.28}$ | $\mathbf{91.84}_{\pm0.34}$ |
| | | 60 | $89.77_{\pm0.55}$ | $88.99_{\pm8.69}$ | $89.15_{\pm0.52}$ | $91.30_{\pm0.28}$ | $72.99_{\pm0.44}$ | $86.54_{\pm0.95}$ | $\underline{87.09}_{\pm0.70}$ | $90.25_{\pm0.29}$ | $\mathbf{91.27}_{\pm0.17}$ | $90.67_{\pm0.33}$ | $\underline{91.39}_{\pm0.14}$ |
| | Mi-F1 | 1 | $51.72_{\pm8.02}$ | $73.71_{\pm5.74}$ | $51.23_{\pm0.76}$ | $84.34_{\pm7.02}$ | $41.84_{\pm1.11}$ | $78.62_{\pm2.53}$ | $\underline{82.83}_{\pm1.63}$ | $89.59_{\pm0.37}$ | $\underline{90.15}_{\pm0.52}$ | $89.71_{\pm0.41}$ | $\mathbf{91.02}_{\pm0.22}$ |
| | | 5 | $83.35_{\pm8.43}$ | $84.03_{\pm3.44}$ | $83.45_{\pm0.89}$ | $83.59_{\pm4.57}$ | $54.82_{\pm0.82}$ | $81.12_{\pm1.20}$ | $\underline{81.85}_{\pm1.89}$ | $91.83_{\pm0.25}$ | $\mathbf{92.12}_{\pm0.42}$ | $91.25_{\pm0.39}$ | $\underline{91.68}_{\pm0.45}$ |
| | | 20 | $90.49_{\pm0.56}$ | $90.29_{\pm2.90}$ | $90.60_{\pm0.54}$ | $90.38_{\pm1.36}$ | $72.49_{\pm0.30}$ | $84.03_{\pm1.20}$ | $\underline{84.41}_{\pm1.32}$ | $91.01_{\pm0.36}$ | $\underline{92.05}_{\pm0.50}$ | $92.16_{\pm0.14}$ | $\mathbf{92.46}_{\pm0.29}$ |
| | | 40 | $90.11_{\pm0.42}$ | $90.85_{\pm5.67}$ | $90.80_{\pm0.47}$ | $90.99_{\pm0.41}$ | $74.56_{\pm0.64}$ | $85.43_{\pm1.09}$ | $\underline{85.91}_{\pm0.91}$ | $91.35_{\pm0.28}$ | $\underline{92.19}_{\pm0.36}$ | $91.72_{\pm0.26}$ | $\mathbf{92.53}_{\pm0.31}$ |
| | | 60 | $91.70_{\pm0.42}$ | $90.25_{\pm6.22}$ | $91.58_{\pm0.48}$ | $92.13_{\pm0.27}$ | $73.63_{\pm0.42}$ | $86.54_{\pm0.95}$ | $\underline{87.09}_{\pm0.70}$ | $91.30_{\pm0.25}$ | $\underline{92.22}_{\pm0.16}$ | $91.80_{\pm0.23}$ | $\mathbf{92.35}_{\pm0.13}$ |
| IMDB | Ma-F1 | 1 | $23.26_{\pm1.59}$ | $28.99_{\pm3.21}$ | $35.75_{\pm1.85}$ | $29.87_{\pm2.28}$ | $31.08_{\pm0.96}$ | $37.70_{\pm2.21}$ | $\underline{40.22}_{\pm2.50}$ | $28.00_{\pm1.65}$ | $\underline{32.51}_{\pm3.86}$ | $38.29_{\pm2.44}$ | $\mathbf{40.28}_{\pm2.83}$ |
| | | 5 | $39.79_{\pm2.21}$ | $35.72_{\pm4.29}$ | $39.59_{\pm1.08}$ | $37.17_{\pm2.79}$ | $37.47_{\pm1.13}$ | $45.58_{\pm3.05}$ | $\underline{49.63}_{\pm1.04}$ | $35.92_{\pm2.60}$ | $\underline{37.66}_{\pm2.28}$ | $48.82_{\pm1.40}$ | $\mathbf{51.87}_{\pm1.69}$ |
| | | 20 | $45.76_{\pm1.87}$ | $48.75_{\pm2.56}$ | $48.77_{\pm0.46}$ | $45.85_{\pm1.62}$ | $44.08_{\pm0.53}$ | $47.30_{\pm5.01}$ | $\underline{49.56}_{\pm1.07}$ | $42.16_{\pm2.17}$ | $\underline{43.75}_{\pm1.43}$ | $50.87_{\pm1.69}$ | $\mathbf{52.14}_{\pm2.27}$ |
| | | 40 | $45.58_{\pm0.78}$ | $47.98_{\pm1.57}$ | $46.37_{\pm0.40}$ | $44.40_{\pm1.73}$ | $42.47_{\pm0.71}$ | $45.25_{\pm3.14}$ | $\underline{48.77}_{\pm1.30}$ | $45.94_{\pm1.74}$ | $\underline{46.48}_{\pm1.50}$ | $51.18_{\pm1.57}$ | $\mathbf{52.81}_{\pm1.36}$ |
| | | 60 | $49.51_{\pm0.72}$ | $51.53_{\pm1.06}$ | $48.97_{\pm0.38}$ | $46.60_{\pm2.30}$ | $44.78_{\pm0.89}$ | $47.14_{\pm7.22}$ | $\underline{51.14}_{\pm1.25}$ | $48.12_{\pm1.27}$ | $\underline{49.19}_{\pm1.42}$ | $52.17_{\pm1.67}$ | $\mathbf{53.83}_{\pm1.36}$ |
| | Mi-F1 | 1 | $38.23_{\pm0.40}$ | $39.33_{\pm1.31}$ | $40.28_{\pm0.96}$ | $37.97_{\pm1.18}$ | $36.16_{\pm1.42}$ | $37.99_{\pm1.85}$ | $\underline{39.95}_{\pm2.51}$ | $33.02_{\pm2.44}$ | $\underline{35.45}_{\pm2.11}$ | $40.19_{\pm1.70}$ | $\mathbf{41.99}_{\pm2.26}$ |
| | | 5 | $42.92_{\pm1.00}$ | $40.25_{\pm1.80}$ | $44.01_{\pm1.48}$ | $39.23_{\pm2.21}$ | $41.54_{\pm0.96}$ | $45.48_{\pm2.99}$ | $\underline{49.39}_{\pm0.98}$ | $37.77_{\pm1.33}$ | $\underline{38.74}_{\pm2.16}$ | $\mathbf{51.77}_{\pm1.17}$ | $51.36_{\pm1.30}$ |
| | | 20 | $45.80_{\pm1.74}$ | $50.29_{\pm2.04}$ | $48.78_{\pm0.42}$ | $46.65_{\pm1.62}$ | $44.85_{\pm0.58}$ | $48.58_{\pm2.99}$ | $\underline{49.22}_{\pm1.12}$ | $42.61_{\pm2.13}$ | $\underline{44.33}_{\pm1.57}$ | $52.08_{\pm1.36}$ | $\mathbf{52.72}_{\pm1.22}$ |
| | | 40 | $45.55_{\pm0.84}$ | $48.68_{\pm1.50}$ | $46.39_{\pm0.35}$ | $44.90_{\pm1.62}$ | $43.36_{\pm0.71}$ | $46.11_{\pm2.65}$ | $\underline{48.52}_{\pm1.31}$ | $46.31_{\pm1.05}$ | $\underline{47.24}_{\pm1.63}$ | $52.14_{\pm1.16}$ | $\mathbf{52.71}_{\pm1.18}$ |
| | | 60 | $49.46_{\pm0.73}$ | $53.05_{\pm0.95}$ | $49.00_{\pm0.41}$ | $47.10_{\pm2.24}$ | $45.52_{\pm0.91}$ | $49.38_{\pm2.90}$ | $\underline{50.86}_{\pm1.31}$ | $48.53_{\pm1.25}$ | $\underline{49.92}_{\pm1.43}$ | $52.41_{\pm1.25}$ | $\mathbf{53.72}_{\pm1.94}$ |

learning rate ranging from 1e-4 to 5e-3. We also tune the patience for early stopping from 20 to 100. The regularization hyperparameter $\lambda$ is set to 0.01. We experiment with the number of feature tokens $K$, searching values from { 1, 5, 10, 15, 20 }. Lastly, for our non-linear activation function $\sigma(\cdot)$, we use LeakyReLU.

## 5.2 Performance on Node Classification (RQ1)

Experiment results for semi-supervised node classification on three benchmark datasets are detailed in Table 2. Compared to the pre-trained DMGI, HeCo, and HDMI models, our post-training prompting framework, HetGPT, exhibits superior performance in 88 out of the 90 comparison pairs. Specifically, we observe a relative improvement of 3.00% in Macro-F1 and 2.62% in Micro-F1. The standard deviation of HetGPT aligns closely with that of the original models, indicating that the improvement achieved is both substantial and robust. It's crucial to note that the three HGNNs with "*pre-train, fine-tune*" - DMGI, HeCo, and HDMI, are already among the state-of-the-art methods for semi-supervised node classification. By integrating them with HetGPT, we push the envelope even further, setting a new performance pinnacle. Furthermore, HetGPT's edge becomes even more significant in scenarios where labeled nodes are extremely scarce, achieving an improvement of 6.60% in Macro-F1 and 6.88% in Micro-F1 under the 1-shot setting. Such

marked improvements in few-shot performance strongly suggest HetGPT's efficacy in mitigating the overfitting issue. The strategic design of our prompting function, especially the virtual class prompt, effectively captures the intricate characteristics of each class, which can potentially obviate the reliance on costly annotated data. Additionally, GPPT lags considerably on all datasets, which further underscores the value of HetGPT's effort in tackling the unique challenges inherent to heterogeneous graphs.

## 5.3 Performance under Different Settings (RQ2)

*5.3.1 Ablation study.* To further demonstrate the effectiveness of each module in HetGPT, we conduct an ablation study to evaluate our full framework against the following three variants:

- **w/o VCP**: the variant of HetGPT without the virtual class prompt from Section 4.2;
- **w/o HFP**: the variant of HetGPT without the heterogeneous feature prompt from Section 4.3;
- **w/o MNA**: the variant of HetGPT without the multi-view neighborhood aggregation from Section 4.4.

Experiment results on ACM and DBLP, shown in Figure 3, highlight the substantial contributions of each module to the overall effectiveness of HetGPT. Notably, the virtual class prompt emerges

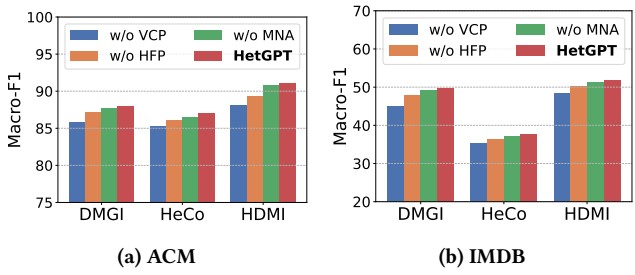

**Figure 3: Ablation study of HetGPT on ACM and IMDB.**

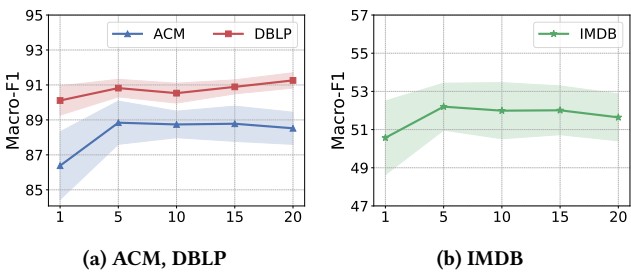

**Figure 4: Performance of HetGPT with the different number of basis feature vectors on ACM, DBLP, and IMDB.**

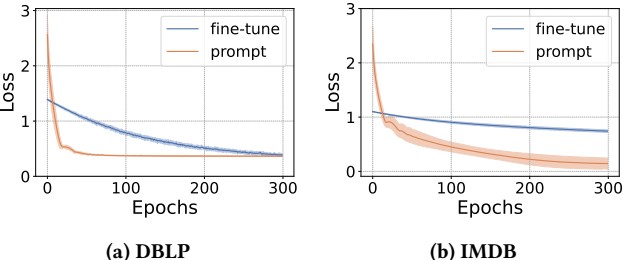

**Figure 5: Comparison of training losses over epochs between HetGPT and its fine-tuning counterpart on DBLP and IMDB.**

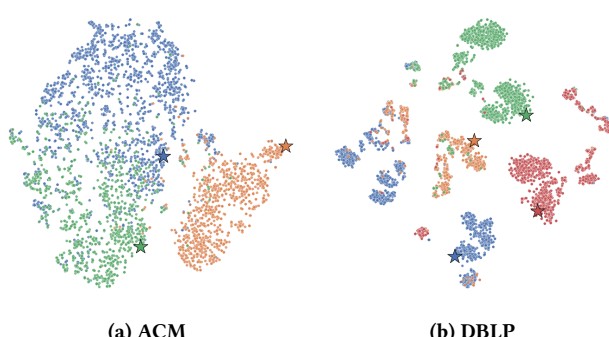

**Figure 6: Visualization of the learned node tokens and class tokens in virtual class prompt on ACM and DBLP.**

as the most pivotal component, indicated by the significant performance drop when it's absent. This degradation mainly stems from the overfitting issue linked to the negative transfer problem, especially when labeled nodes are sparse. The virtual class prompt directly addresses this issue by generalizing the intricate characteristics of each class within the embedding space.

*5.3.2 Hyper-parameter sensitivity.* We evaluate the sensitivity of HetGPT to its primary hyperparameter: the number of basis feature tokens $K$ in Equation (7). As depicted in Figure 4, even a really small value of $K$ (*i.e.,* 5 for ACM, 20 for DBLP, and 5 for IMDB) can lead to satisfactory node classification performance. This suggests that the prompt tuning effectively optimizes performance without the need to introduce an extensive number of new parameters.

## 5.4 Prompt Tuning Efficiency Analysis (RQ3)

Our HetGPT, encompassing the virtual class prompt and the heterogeneous feature prompt, adds only a few new trainable parameters (*i.e.,* comparable to a shallow MLP). Concurrently, the parameters of the pre-trained HGNNs and the contrastive head remain unchanged during the entire prompt tuning phase. Figure 5 illustrates that HetGPT converges notably faster than its traditional "*pre-train, fine-tune*" counterpart, both recalibrating the parameters of the pre-trained HGNNs and introducing a new prediction head. This further demonstrates the efficiency benefits of our proposed framework, allowing for effective training with minimal tuning iterations.

## 5.5 Interpretability Analysis (RQ4)

To gain a clear understanding of how the design of the virtual class prompt facilitates effective node classification without relying on the traditional classification paradigm, we employ a t-SNE plot to visualize the node representations and the learned virtual class

prompt on ACM and DBLP, as shown in Figure 6. Within this visualization, nodes are depicted as colored circles, while the class tokens from the learned virtual class prompt are denoted by colored stars. Each color represents a unique class label. Notably, the embeddings of these class tokens are positioned in close vicinity to clusters of node embeddings sharing the same class label. This immediate spatial proximity between a node and its respective class token validates the efficacy of similarity measures inherited from the contrastive pretext for the downstream node classification task. This observation further reinforces the rationale behind our node classification approach using the virtual class prompt, *i.e.,* a node is labeled as the class that its embedding is most closely aligned with.

## 6 CONCLUSION

In this paper, we propose HetGPT, a general post-training prompting framework to improve the node classification performance of pre-trained heterogeneous graph neural networks. Recognizing the prevalent issue of misalignment between the objectives of pretext and downstream tasks, we craft a novel prompting function that integrates a virtual class prompt and a heterogeneous feature prompt. Furthermore, our framework incorporates a multi-view neighborhood aggregation mechanism to capture the complex neighborhood structure in heterogeneous graphs. Extensive experiments on three benchmark datasets demonstrate the effectiveness of HetGPT. For future work, we are interested in exploring the potential of prompting methods in tackling the class-imbalance problem on graphs or broadening the applicability of our framework to diverse graph tasks, such as link prediction and graph classification.

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
