# OpenReview forum: "HetGPT: Harnessing the Power of Prompt Tuning in Pre-Trained Heterogeneous Graph Neural Networks"
_ACM.org/TheWebConf/2024/Conference — TheWebConf24_

### Official Review · Reviewer_7Adk · 2023-11-23

**Novelty:** 6
**Technical Quality:** 4

**Review:**

This paper proposes HetGPT, a novel post-training prompting framework specifically designed for heterogeneous graphs.
Given the inherent misalignment between training objectives of pretext tasks and downstream tasks in graph machine learning, especially in scenarios with limited labeled nodes, HetGPT aims to reformulate downstream tasks to better align with contrastive pretext tasks.
The HetGPT model integrates a virtual class prompt and a heterogeneous feature prompt, alongside a multi-view neighborhood aggregation mechanism. This approach effectively bridges the gap between pre-training objectives and downstream tasks on heterogeneous graphs.
Extensive experiments conducted on three benchmark datasets show that HetGPT enhances the performance of state-of-the-art heterogeneous graph neural networks (HGNNs) in semi-supervised node classification tasks​​​​.

### Pros:
**Innovative Approach for Heterogeneous Graphs:** HetGPT's integration of a virtual class prompt and a heterogeneous feature prompt is a novel approach in the realm of heterogeneous graph neural networks.
**Enhanced Performance:** Demonstrates improved performance over state-of-the-art HGNNs in semi-supervised node classification, validated across multiple benchmark datasets.
**Clear and Coherent Writing**:  The paper articulates its motivation and preliminary concepts in an easily understandable manner, enhancing its accessibility to readers.

### Cons:

**Limited Interpretation of Prompt-based Methods**: While HetGPT effectively aligns pre-training and downstream tasks, it seems to overlook a critical aspect of prompt-based methods prevalent in NLP - leveraging inherent knowledge or common sense encoded in the data. In NLP, prompts like "I feel so [mask]" are used to evoke specific emotional responses based on common understanding. The framework primarily focuses on task alignment but may not fully exploit the inherent knowledge embedded within graph structures, which is a key aspect of prompt-based learning.

**Questions:**

1. Can the authors provide a case study that intuitively demonstrates the effectiveness of the virtual class prompt in HetGPT? While the verification through t-SNE visualizations offers some insights, an in-depth case study could more clearly illustrate how the virtual class prompt in the feature space influences model performance and decision-making processes. This would help in understanding the practical implications and real-world applications of HetGPT.

**Reviewer Confidence:**

2: The reviewer is willing to defend the evaluation, but it is likely that the reviewer did not understand parts of the paper

**Scope:**

4: The work is relevant to the Web and to the track, and is of broad interest to the community

---

### Official Review · Reviewer_irbr · 2023-11-23

**Novelty:** 5
**Technical Quality:** 5

**Review:**

The paper introduces a general post-training prompting framework, called HetGPT, which is designed specifically for heterogeneous graphs. The paper also proposes a novel prompting function that integrates a virtual class prompt with a heterogeneous feature prompt.

Strength:
The paper adapts the “pre-train, prompt” paradigm to heterogeneous graphs for the first time. It is well presented and structured. The experiments are extensive to certify the effectiveness of HetGPT on semi-supervised node classification task.

Weakness:
While the paper claims to propose a general post-training prompting framework for HGNNs, certain aspects the method are only specifically designed for node classification. And the paper only evaluates the semi-supervised node classification task in experiments.

**Questions:**

See the weakness.

**Reviewer Confidence:**

3: The reviewer is confident but not certain that the evaluation is correct

**Scope:**

4: The work is relevant to the Web and to the track, and is of broad interest to the community

---

### Official Review · Reviewer_mHY5 · 2023-11-23

**Novelty:** 5
**Technical Quality:** 4

**Review:**

The paper introduces HetGPT, a post-training prompting framework designed for heterogeneous graphs. The framework includes a novel graph prompting function with a virtual class prompt and a heterogeneous feature prompt, aiming to align downstream tasks with pretext tasks. Additionally, a multi-view neighborhood aggregation mechanism captures the complex neighborhood structure in heterogeneous graphs. Experiments on three benchmark datasets demonstrate the effectiveness of the HetGPT in comparison with heterogeneous graph neural network baselines.

**Strengths:**

(+) The idea of using the "pre-train, prompt" paradigm in graph neural networks is an important and popular topic, inspired by its success in natural language processing. Extending the idea to heterogeneous networks is well-motivated.

(+) The presented techniques, including the heterogeneous feature prompt and the multi-view neighborhood aggregation mechanism, are reasonable and intuitive given the network heterogeneity.

(+) Various heterogeneous GNN baselines, including supervised, generative, contrastive, and "pre-train, prompt" ones are compared. The authors also conduct ablation studies and hyperparameter analysis to validate the contribution of their presented techniques and configurations.

**Weaknesses:**

(-) The framework design (e.g., the virtual class prompt) is centered around the node classification task only. The experiments also consider semi-supervised node classification only. It is unclear if the effectiveness of HetGPT can be generalized to other tasks (e.g., link prediction). In fact, the goal of "pre-train, prompt" is to make the model capable of multiple tasks (or at least classification in different label spaces).

(-) Statistical significance tests are missing. It is unclear whether the gaps between HetGPT and baselines/ablation versions are statistically significant or not. In fact, in Table 2 and Figure 3, some gaps are quite subtle, therefore p-values should be reported.

**Questions:**

- Could you explain how the idea of "pre-train, prompt" can be extended to other tasks (e.g., link prediction) in (heterogeneous) graphs?

- Could you show the generalizability of HetGPT by performing classification in different label spaces and/or for different types of nodes?

- Could you conduct statistical significance tests to compare HetGPT with the strongest baseline in Table 2 and each ablation version in Figure 3?

- Could you provide a theoretical analysis of the number of trainable parameters in HetGPT? This can echo the empirical efficiency analysis in Section 5.4.

**Reviewer Confidence:**

3: The reviewer is confident but not certain that the evaluation is correct

**Scope:**

3: The work is somewhat relevant to the Web and to the track, and is of narrow interest to a sub-community

---

### Official Review · Reviewer_G3Ba · 2023-11-25

**Novelty:** 5
**Technical Quality:** 5

**Review:**

This is an interesting paper on using pre-trained models on heterogenous graphs, with learning and fine tuning of prompts with a small number of labeled examples. At times, the writing felt unpolished, in the sense that while the subsections were reasonably well written, how the pieces fit together was not always clear. The paper would benefit from a more careful editing pass to ensure that the flow of the paper is smooth and the connections between the different parts are clear; to their credit, the authors attempted to do this in figure 2.

While reading the paper, it also felt like the authors were trying to do too many things at the same time to demonstrate the effectiveness of their approach. For example, the authors discuss multi-view aggregation in section 4.4; while this is a reasonable thing to do to boost performance, it distracts from the main point of the paper, which is to show that prompts can be used to fine-tune pre-trained models on heterogenous graphs.  The authors could have instead performed a more detailed analysis of (section 4.3) in their paper especially in the few shot setting, where they have the best results.

The authors show that the best results occur in the few-shot setting (e.g., the number of examples is five or less). But we don't understand why this is the case. What aspects of their work has led to this result? It would be great to break down these results (i.e. diminishing returns with more examples) and break them down in relation to the different aspects of their work. As a secondary point, the gains in performance don't appear to be statistically significant; the distributions of HetGPT+ and the baselines overlap.

It would have been a more robust experimental setting had the authors used a graph from a different domain with the same types and labels as the original graph (e.g., two different citation graphs) for the transfer learning experiments (i.e., pre-train on citation graph 1 and predict on graph 2 with prompt fine-tuning).

The number of classes is quite small (3--4); it would be interesting to see how the approach performs with a larger number of classes, possibly with a skewed class distribution.

**Questions:**

* Why does the performance gain with $K$ the number of basis vectors stabilize with $K \ge 5$? Is there a theoretical reason for this?
* At the moment, it is not clear how much of the prompt tuning benefit is related to the similarity in the local graph structure distribution between the pre-training and fine-tuning graphs since they are from the same graph dataset (i.e., would we see a drop in the gains if there was a distribution shift?).
* It appears like the virtual class prompt acts a regularizer. But the authors don't provide class statistics, that is, are all classes equally likely? What if the class distribution is skewed? Would the virtual class prompt still be effective?

**Ethics Review Description:**

no concerns

**Reviewer Confidence:**

4: The reviewer is certain that the evaluation is correct and very familiar with the relevant literature

**Scope:**

4: The work is relevant to the Web and to the track, and is of broad interest to the community

---

### Decision · Program_Chairs · 2024-01-22

**Decision:**

Accept

**Comment:**

This submission studies prompt tuning for graph representation learning. It introduces a post training prompting strategy for GNNs pre-trained on heterogeneous networks. Its technical contribution lies in the idea of "pre-train, prompt" for GNNs, certainly, inspired from the LLM paradigm in NLP. Experiments are performed on several (small-scale) datasets with interesting results.

 The strength of this work lies in the prompting idea for pre-trained heterogeneous graph neural nets, as identified by almost all reviewers.

 Some issues raised by the reviewers include 1) the gains are marginal and thus statistical tests are suggested --- addressed during rebuttal; 2) it focuses only on the node classification task and link prediction is suggested --- explained during rebuttal; and 3) the organization and presentation of certain parts requires improvements --- promised to address during rebuttal.

 Overall, it receives ratings of 5, 5, 5, 6 in terms of novelty and 5, 4, 5, 4 in terms of technical quality, suggesting the reviewers in favor of this work. Scope-wise, it is relevant to the Web and the graph track.

 Thus, this is a relatively easy decision to recommend an acceptance for this submission. Note that the authors are suggested to address the issues and perform improvements as promised during the rebuttal phase.